# One Million Scenes for Autonomous Driving: ONCE Dataset

**Jiageng Mao** [1*]       **Minzhe Niu** [2*]       **Chenhan Jiang** [2]       **Hanxue Liang** [5]

**Jingheng Chen** [3]   **Xiaodan Liang** [4†]   **Yamin Li** [3]   **Chaoqiang Ye** [2]   **Wei Zhang** [2]

**Zhenguo Li** [2]       **Jie Yu** [3]       **Hang Xu** [2†]       **Chunjing Xu** [2]

## Abstract

Current perception models in autonomous driving have become notorious for greatly relying on a mass of annotated data to cover unseen cases and address the long-tail problem. On the other hand, learning from unlabeled large-scale collected data and incrementally self-training powerful recognition models have received increasing attention and may become the solutions of next-generation industry-level powerful and robust perception models in autonomous driving. However, the research community generally suffered from data inadequacy of those essential real-world scene data, which hampers the future exploration of fully/semi/self-supervised methods for 3D perception. In this paper, we introduce the ONCE (**O**ne millio**N** s**C**en**E**s) dataset for 3D object detection in the autonomous driving scenario. The ONCE dataset consists of 1 million LiDAR scenes and 7 million corresponding camera images. The data is selected from 144 driving hours, which is 20x longer than the largest 3D autonomous driving dataset available (*e.g.* nuScenes and Waymo), and it is collected across a range of different areas, periods and weather conditions. To facilitate future research on exploiting unlabeled data for 3D detection, we additionally provide a benchmark in which we reproduce and evaluate a variety of self-supervised and semi-supervised methods on the ONCE dataset. We conduct extensive analyses on those methods and provide valuable observations on their performance related to the scale of used data. Data, code, and more information are available at http://www.once-for-auto-driving.com.

## 1 Introduction

Autonomous driving is a promising technology that has the potential to ease the drivers' burden and save human lives from accidents. In autonomous driving systems, 3D object detection is a crucial technique that can identify and localize the vehicles and humans surrounding the self-driving vehicle, given 3D point clouds from LiDAR sensors and 2D images from cameras as input. Recent advances [4, 39] in 3D object detection demonstrate that large-scale and diverse scene data can significantly improve the perception accuracy of 3D detectors.

Unlike other image-based datasets (*e.g.* ImageNet [12], MS COCO [26]) in which the training data can be obtained directly from websites and the annotation pipeline is relatively simple, the research community generally faces two major problems on the acquisition and exploitation of scene data for autonomous driving: 1) The data resources are scarce and the scenes generally lack diversity. The scenes for autonomous driving must be collected by driving a car that carries an expensive sensor suite

---

  [*] Equal contribution.        [1] The Chinese University of Hong Kong        [2] Huawei Noah's Ark Lab
  [3] Huawei IAS BU Vehicle Cloud Service        [4] Sun Yat-Sen University        [5] ETH Zurich
  [†] Corresponding authors: `xu.hang@huawei.com` & `xdliang328@gmail.com`

35th Conference on Neural Information Processing Systems (NeurIPS 2021) Track on Datasets and Benchmarks.

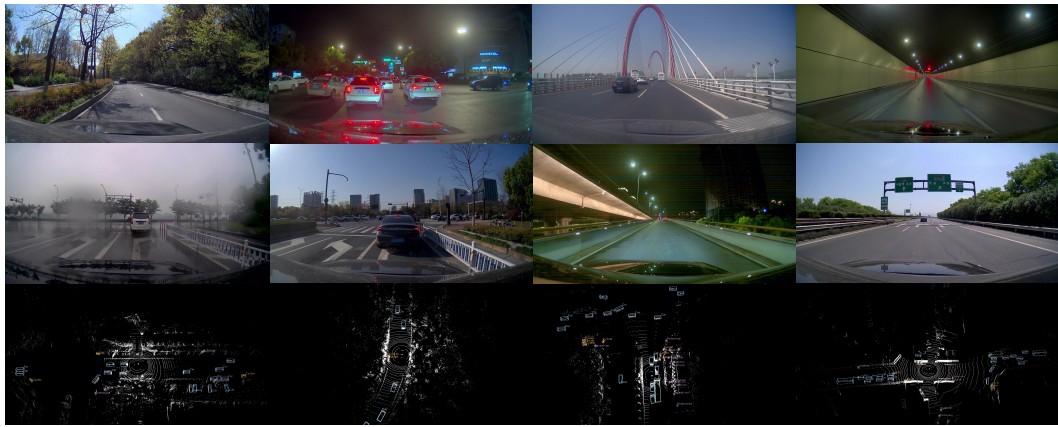

Figure 1: Images and point clouds sampled from the ONCE (**O**ne millio**N** s**C**en**E**s) dataset. Our ONCE dataset covers a variety of geographical locations, time periods and weather conditions.

on the roads in compliance with local regulations. Thus existing autonomous driving datasets could only provide a limited amount of scene data. For instance, on the largest Waymo Open dataset [39], the scene data is recorded with only 6.4 driving hours, which can hardly cover enough different circumstances. 2) Effectively leveraging unlabeled scene data becomes an important problem in practical applications. Typically a data acquisition vehicle can collect more than 200k frames of point clouds with 8 working hours, but a skilled worker can only annotate 100-200 frames per day. This will lead to a rapid accumulation of a large amount of unlabeled data. Although algorithms of semi-supervised learning [32, 40, 45], self-supervised learning [18, 17] and unsupervised domain adaptation [27, 14] show promising results to handle those unlabeled data on the image domain, currently only a few methods [42, 46] are studied for the autonomous driving scenario, mainly because of the limited data amount provided by existing datasets.

To resolve the data inadequacy problem, in this paper, we introduce the ONCE (**O**ne millio**N** s**C**en**E**s) dataset, which is the largest and most diverse autonomous driving dataset to date. The ONCE dataset contains 1 million 3D scenes and 7 million corresponding 2D images, which is 5x quantitatively more than the largest Waymo Open dataset [39], and the 3D scenes are recorded with 144 driving hours, which is 20x longer and covers more diverse weather conditions, traffic conditions, time periods and areas than existing datasets. Figure 1 shows various scenes in the ONCE dataset. Each scene is captured by a high-quality LiDAR sensor and transformed to dense 3D point clouds with 70k points per scene in average. For each scene, 7 cameras capture high-resolution images that cover 360° field of view. The data of LiDAR sensor and cameras are precisely synchronized and additional calibration and transformation parameters are provided to enable the fusion of multiple sensors and scenes. We exhaustively annotated 16k scenes with 3D ground truth boxes of 5 categories (car, bus, truck, pedestrian and cyclist), giving rise to 417k 3D boxes in total. And 769k 2D bounding boxes are also provided for camera images by projecting 3D boxes into image planes. The other scenes are kept unannotated, mainly to facilitate future research on the exploitation of unlabeled data. Comprehensive comparisons between the ONCE dataset and other autonomous driving datasets are in Table 1.

To resolve the unlabeled data exploitation problem and facilitate future research on this area, in this paper, we introduce a 3D object detection benchmark in which we implement and evaluate a variety of self-supervised and semi-supervised learning methods on the ONCE dataset. Specifically, we first carefully select a bunch of widely-used self-supervised and semi-supervised learning methods, including classic image algorithms and methods for the indoor 3D scenario. Then we adapt those methods to the task of 3D object detection for autonomous driving and reproduce their methods with the same detection framework. We train and evaluate those approaches and finally provide some observations on semi-/self-supervised learning for 3D detection by analyzing the obtained results. We also provide baseline results for multiple 3D detectors and unsupervised domain adaptation methods. Extensive experiments show that models pretrained on the ONCE dataset perform much better than those pretrained on other datasets (nuScenes and Waymo) using the same self-supervised method, which implies the superior data quality and diversity of our dataset.

| Dataset | Scenes | Size (hr.) | Area (km²) | Images | 3D boxes | night/ rain | Cls. |
|---|---|---|---|---|---|---|---|
| KITTI [15] | 15k | 1.5 | - | 15k | 80k | No/No | 3 |
| ApolloScape [28] | 20k | 2 | - | 0 | 475k | No/No | 6 |
| KAIST [11] | 8.9k | - | - | 8.9k | 0 | Yes/No | 3 |
| A2D2 [16] | 40k | - | - | - | - | No/Yes | 14 |
| H3D [31] | 27k | 0.8 | - | 83k | 1.1M | No/No | 8 |
| Cityscapes 3D [13] | 20k | - | - | 20k | - | No/No | 8 |
| Argoverse [7] | 44k | 1 | 1.6 | 490k | 993k | Yes/Yes | 15 |
| Lyft L5 [20] | 30k | 2.5 | - | - | 1.3M | No/No | 9 |
| A*3D [33] | 39k | 55 | - | 39k | 230k | Yes/Yes | 7 |
| nuScenes [4] | 400k | 5.5 | 5 | 1.4M | 1.4M | Yes/Yes | 23 |
| Waymo Open [39] | 230k | 6.4 | 76 | 1M | 12M | Yes/Yes | 4 |
| ONCE (ours) | **1M** | **144** | **210** | **7M** | 417k | **Yes/Yes** | 5 |

Table 1: Comparisons with other 3D autonomous driving datasets. "-" means not mentioned. Our ONCE dataset has 4x scenes, 7x images, and 20x driving hours compared with the largest dataset [39].

Our main contributions can be summarized into two folds: 1) We introduce the ONCE dataset, which is the largest and most diverse autonomous driving dataset up to now. 2) We introduce a benchmark of self-supervised and semi-supervised learning for 3D detection in the autonomous driving scenario.

## 2 Related Work

**Autonomous driving datasets.** Most autonomous driving datasets collect data on the roads with multiple sensors mounted on a vehicle, and the obtained point clouds and images are further annotated for perception tasks including detection and tracking. The KITTI dataset [15] is a pioneering work in which they record 22 road sequences with stereo cameras and a LiDAR sensor. The ApolloScape dataset [19] offers per-pixel semantic annotations for 140k camera images and [28] additionally provides point cloud data based on the ApolloScape. The KAIST Multi-Spectral dataset [11] uses thermal imaging cameras to record scenes. The H3D dataset [31] provides point cloud data in 160 urban scenes. The Argoverse dataset [7] introduces geometric and semantic maps. The Lyft L5 dataset [20] and the A*3D dataset [33] offer 46k and 39k annotated LiDAR frames respectively.

The nuScenes dataset [4] and the Waymo Open dataset [39] are currently the most widely-used autonomous driving datasets. The nuScenes dataset records 5.5 hours driving data by multiple sensors with 400k 3D scenes in total, and the Waymo Open dataset offers 200k scenes of 6.4 driving hours with massive annotations. Compared with those two datasets, our ONCE dataset is not only quantitatively larger in terms of scenes and images, *e.g.* 1M scenes versus 200k in [39], but also more diverse since our 144 driving hours cover all time periods as well as most weather conditions. Statistical comparisons with other autonomous driving datasets are shown in Table 1.

**3D object detection in driving scenarios.** Many techniques have been explored for 3D object detection in driving scenarios, and they can be broadly categorized into two classes: 1) Single-modality 3D detectors [47, 35, 23, 36, 50, 56, 34, 48] are designed to detect objects solely from sparse point clouds. PointRCNN [35] operates directly on point clouds to predict bounding boxes. SECOND [47] rasterizes point clouds into voxels and applies 3D convolutional networks on voxel features to generate predictions. PointPillars [23] introduces the pillar representation to project point clouds to Bird Eye View (BEV) and utilizes 2D convolutional networks for object detection. PV-RCNN [36] combines point clouds and voxels for proposal generation and refinement. CenterPoints [50] introduces the center-based assignment scheme for accurate object localization. 2) Multi-modality approaches [41, 9, 21, 37, 25, 51] leverage both point clouds and images for 3D object detection. PointPainting [41] uses images to generate segmentation maps and appends the segmentation scores to corresponding point clouds to enhance point features. Other methods [9, 21] try to fuse point features and image features on multiple stages of a detector. Our benchmark evaluates a variety of 3D detection models, including both single-modality and multi-modality detectors on the ONCE dataset.

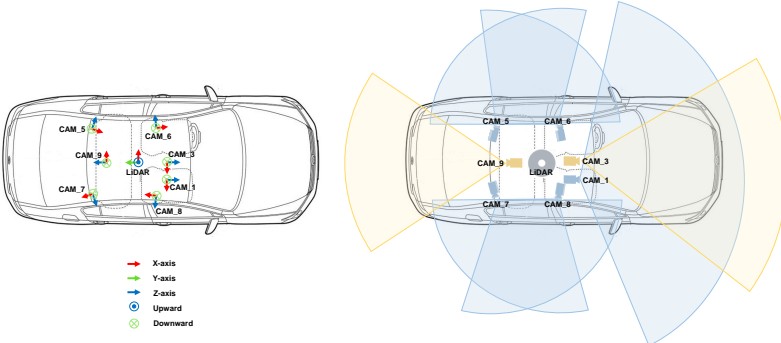

Figure 2: Sensor locations and coordinate systems. The data acquisition vehicle is equipped with 1 LiDAR and 7 cameras that can capture 3D point clouds and images from 360° field of view.

| | Freq. (Hz) | HFOV (°) | VFOV (°) | Size | Range (m) | Accuracy (cm) | Points/second |
|---|---|---|---|---|---|---|---|
| CAM_1,9 | 10 | 60.6 | [-18, +18] | 1920×1020 | n/a | n/a | n/a |
| CAM_3-8 | 10 | 120 | [-37, +37] | 1920×1020 | n/a | n/a | n/a |
| LiDAR | 10 | 360 | [-25, +15] | n/a | [0.3, 200] | ±2 | 7.2M |

Table 2: Detailed parameters of LiDAR and cameras.

**Deep learning on unlabeled data.** Semi-supervised learning and self-supervised learning are two promising areas in which various emerging methods are proposed to learn from the unlabeled data effectively. Methods on semi-supervised learning are mainly of two branches: The first branch of methods try to annotate those unlabeled data with pseudo labels [24, 3, 2, 32] by self-training [45] or teacher model [40]. Other methods [44, 27, 22, 29, 38] regularize pairs of augmented images under consistency constraints. Self-supervised learning approaches learn from the unlabeled data by leveraging auxiliary tasks [53, 30] or by clustering [5, 6, 1]. Recent advances [18, 17, 10, 8] demonstrate that contrastive learning methods show promising results in self-supervised learning. Semi-/self-supervised learning has also been studied in 3D scenarios. SESS [55] is a semi-supervised method that utilizes geometric and semantic consistency for indoor 3D object detection. 3DIoUMatch [42] utilizes an auxiliary IoU head for boxes filtering. For self-supervised learning, PointContrast [46] and DepthContrast [54] apply contrastive learning on point clouds. Our benchmark provides a fair comparison of various self-supervised and semi-supervised methods.

## 3 ONCE Dataset

### 3.1 Data Acquisition System

**Sensor specifications.** The data acquisition system is built with one 40-beam LiDAR sensor and seven high-resolution cameras mounted on a car. The specific sensor layout is shown in Figure 2, and the detailed parameters of all sensors are listed in Table 2. We note that the LiDAR sensor and the set of cameras can both capture data covering 360° horizontal field of view near the driving vehicle, and all the sensors are well-synchronized, which enables good alignments of cross-modality data. We carefully calibrate the intrinsics and extrinsics of each camera using calibration target boards with patterns. We check the calibration parameters every day and re-calibrate the sensor that has errors. We make the intrinsics and extrinsics public along with the data to users for camera projection.

**Data collection and annotation.** The data was collected in multiple cities in China. We conform to the local regulations and avoid releasing specific city names and locations. The data collection process lasts 3 months. 3D ground truth boxes were manually labeled from point clouds by annotators

using a commercial annotation system. The labeled boxes then went through a double-check process for validness and refinement, which guarantees high-quality bounding boxes for 3D object detection.

**Data protection.** The driving scenes are collected in permitted areas. We comply with the local regulations and avoid releasing any localization information including GPS information and map data. For privacy protection, we actively detect any object on each image that may contain personal information, *e.g.* human faces, license plates, with a high recall rate, and then we blur those detected objects to ensure no personal information is disclosed.

### 3.2   Data Format

**Coordinate systems.** There are 3 types of coordinate systems on the ONCE dataset, *i.e.*, the LiDAR coordinate, the camera coordinates, and the image coordinate. The LiDAR coordinate is placed at the center of the LiDAR sensor, with the x-axis positive to the left, the y-axis positive backwards, and the z-axis positive upwards. We additionally provide a transformation matrix (vehicle pose) between current frame and the first frame, which enables the fusion of multiple point clouds. The camera coordinates are placed at the center of the lens respectively, with the x-y plane parallel to the image plane and the z-axis positive forwards. The camera coordinates can be transformed to the LiDAR coordinate directly using the respective camera extrinsics. The image coordinate is a 2D coordinate system where the origin is at the top-left of the image, and the x-axis and the y-axis are along the image width and height respectively. The camera intrinsics enable the projection from the camera coordinate to the image coordinate. An illustration of our coordinate systems is in Figure 2.

**LiDAR data.** The original LiDAR data is recorded at a speed of 10 frames per second (FPS). We further downsample those original data with the sampling rate of 2 FPS, since most adjacent frames are quite similar thus redundant. The downsampled data is then transformed into 3D point clouds, resulting in 1 million point clouds, *i.e.*, scenes in total. Each point cloud is represented as an $N \times 4$ matrix, where $N$ is the number of points in this scene, and each point is a 4-dim vector (x, y, z, r). The 3D coordinate (x, y, z) is based on the LiDAR coordinate, and r denotes the reflection intensity. The point clouds are stored into separate binary files for each scene and can be easily read by users.

**Camera data.** The camera data is also downsampled along with the LiDAR data for synchronization, and then the distortions are removed to enhance the quality of the images. We finally provide JPEG compressed images for all the cameras, resulting in 7 million images in total.

**Annotation format.** We select 16k most representative scenes and exhaustively annotate all the 3D bounding boxes of 5 categories: car, bus, truck, pedestrian and cyclist. Each bounding box is a 3D cuboid and can be represented as a 7-dim vector: (cx, cy, cz, l, w, h, $\theta$), where (cx, cy, cz) is the center of the cuboid on the LiDAR coordinate, (l, w, h) denotes length, width, height, and $\theta$ is the yaw angle of the cuboid. We provide 2D bounding boxes by projecting 3D boxes on image planes.

**Other information.** Weather and time information of each scene is useful since it contains explicit domain knowledge, but existing datasets seldom release those important data. In the ONCE dataset, we provide 3 weather conditions, *i.e.*, sunny, cloudy, rainy, and 4 time periods, *i.e.*, morning, noon, afternoon, night, for every labeled and unlabeled scene. We pack all the information, *i.e.*, weather, period, timestamp, pose, calibration, annotations, into a single JSON file for each scene.

**Dataset splits.** The ONCE dataset contains 581 sequences in total. We carefully select and annotate 6 sequences (5k scenes) captured in sunny days as the training split, 4 sequences (3k scenes in total. 1 sequence collected in a sunny day, 1 in a rainy day, 1 in a sunny night and 1 in a rainy night) as the validation split, 10 sequences (8k scenes in total. 3 sequences in sunny days, 3 in rainy days, 2 in sunny nights and 2 in rainy nights) as the testing split. We note that the sequences in each split can cover both downtown and suburban areas. The validation and testing split have quite similar data distributions, and the training split has a slight domain shift compared to the validation/testing split. We choose this way mainly to encourage the proposed methods to have better generalizability.

One major goal of the ONCE dataset is to facilitate research on leveraging large-scale unlabeled data. Thus we keep the remaining 560 sequences as unlabeled, and those sequences can be used for semi-supervised and self-supervised 3D object detection. To explore the effects of different data amounts used for 3D detection, we also divide the unlabeled scenes into 3 subsets: $U_{small}$, $U_{medium}$ and $U_{large}$. The small unlabeled set $U_{small}$ contains 70 sequences (100k scenes), the medium set $U_{medium}$ contains 321 sequences (500k scenes) and the large set $U_{large}$ contains 560 sequences

| pretrain / downstream | KITTI (moderate mAP) | nuScenes (NDS) | Waymo (L2 mAP) |
|---|---|---|---|
| nuScenes→ | 66.1 | - | 53.9 |
| Waymo→ | 66.5 | 49.9 | - |
| ONCE→ | **67.2** | **51.5** | **54.4** |

Table 3: Quality analysis. The model pretrained on the ONCE dataset shows superior performance compared to those on the nuScenes and Waymo dataset, which implies our superior data quality.

| | Time | Weather | Area |
|---|---|---|---|
| Waymo [39] | daytime (80.79%); dawn (9.04%); night (10.17%) | sunny (99.40%); rainy (0.60%) | only city-level labels |
| nuScenes [4] | daytime (88.32%); night (11.68%) | sunny (80.47%); rainy (19.53%) | only city-level labels |
| BDD100k [52] | daytime (52.57%); dawn (7.27%); night (39.94%); undefined (0.22%) | clear (53.45%); overcast (12.53%); partly cloudy (7.04%); rainy (7.27%); snowy (7.91%); foggy (0.18%); undefined (11.61%) | city street (62.14%); highway (24.89%); residential (11.68%); parking lot (0.53%); tunnel (0.20%); gas stations (0.04%); undefined (0.52%) |
| ONCE | morning (39.34%); noon (3.76%); afternoon (36.67%); night (20.24%) | sunny (63.8%); cloudy (30.09%); rainy (6.11%) | downtown (34.29%); suburbs (50.98%); tunnel (1.83%); highway (11.87%); bridge (1.02%) |

Table 4: Diversity analysis. Time and weather labels on nuScenes and Waymo are extracted from scene descriptions and annotations respectively. Both two datasets only provide city-level labels instead of specific road types. Compared to other datasets, our ONCE dataset contains sufficient data captured in rainy days and at nights, and we additionally provide the label of road type for each scene.

(about 1M scenes) in total. We note that $U_{small} \subset U_{medium} \subset U_{large}$ and $U_{small}$, $U_{medium}$ are created by selecting particular roads in time order instead of uniformly sampling from all the scenes, which is more practical since the driving data is usually incrementally updated in real applications.

### 3.3  Dataset Analysis

**Quality analysis.** In order to evaluate the data quality and provide a fair comparison across different datasets, we propose an approach that utilizes pretrained models to imply the respective data quality. Specifically, we first pretrain 3 same backbones of the SECOND detector [47] by the self-supervised method DeepCluster [5] using data from nuScenes [4], Waymo [39] and ONCE respectively, and then we finetune those pretrained models on multiple downstream datasets under the same settings and report their performances. The superior model should have the best pretrained backbone, which means its corresponding pretraining dataset has the best data quality. Table 3 shows the final results. The model pretrained on the ONCE dataset attains 67.2% moderate mAP on the downstream KITTI [15] dataset, and significantly outperforms models pretrained on the Waymo dataset (66.5%) and nuScenes dataset (66.1%), which implies our superior data quality compared to nuScenes and Waymo.

**Diversity analysis.** We analyze the ratios of different weather conditions, time periods and areas in the ONCE dataset and compare them to those in other datasets in Table 4. Our ONCE dataset contains more rainy scenes which accounts for 20% of the total scenes, compared to 10% in Waymo and 12% in nuScenes. The ONCE dataset also provides a sufficient amount of data captured at night with 6% of the total scenes. It is worth noting that we provide explicit labels of both time, weather,

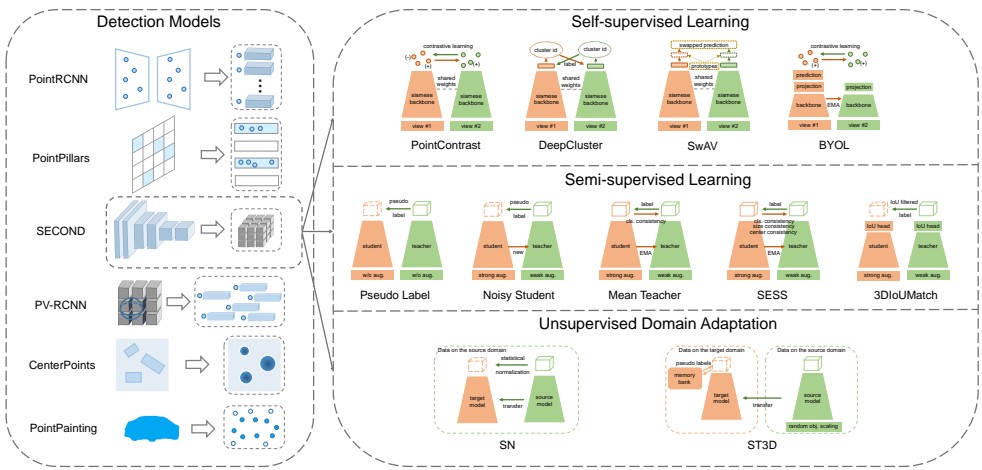

Figure 3: An overview of our 3D object detection benchmark. We reproduce 6 detection models, 4 self-supervised learning, 5 semi-supervised learning, and 2 unsupervised domain adaptation methods for 3D object detection. We give comprehensive analyses on the results and offer valuable observations.

and area for each scene, while Waymo and nuScenes didn't provide labels of collecting areas. The driving scenes can cover most road types including downtown, suburb, highway, bridge and tunnel.

## 4 Benchmark for 3D Object Detection

In this section, we present a 3D object detection benchmark on the ONCE dataset. We reproduce widely-used detection models as well as various methods of self-supervised learning, semi-supervised learning and unsupervised domain adaptation on 3D object detection. We validate those methods with a unified standard and provide performance analysis as well as suggestions for future research.

### 4.1 Models for 3D Object Detection

We implement 5 most widely-used single-modality 3D detectors: PointRCNN [35], PointPillars [23], SECOND [47], PV-RCNN [36] and CenterPoints [50] using only point clouds as input, as well as 1 multi-modality detector PointPainting [41] using both point clouds and images as input on the ONCE dataset. We train those detectors on the training split and report the overall and distance-wise $AP_{3D}^{Ori}$ on the testing split using the evaluation metric in appendix E. Performance on the validation split is also reported in appendix B. We provide training and implementation details in appendix C.

**Points vs. voxels vs. pillars.** Multiple representations (points/voxels/pillars) have been explored for 3D detection. Our experiments in Table 5 demonstrate that the point-based detector [35] performs poorly with only 31.8% mAP on the ONCE dataset, since small objects like pedestrians are naturally sparse and a small amount of used points cannot guarantee a high recall rate. The voxel-based detector [47] shows decent performance with 51.90% mAP compared with 45.47% mAP of the pillar-based detector [23]. It is mainly because voxels contain finer geometric information than pillars. PV-RCNN [36] combines both the point representation and the voxel representation and further improves the detection performance to 53.85% mAP.

**Anchor assignments vs. center assignments.** The only difference between SECOND [47] and CenterPoints [50] is that SECOND uses anchor-based target assignments while CenterPoints introduces the center-based assignments. SECOND shows better performance on the vehicle category (69.71% vs. 66.35%) while CenterPoints performs much better on the small objects including pedestrians (51.80% vs. 26.09%) and cyclists (65.57% vs. 59.92%) in our experiments. It is because the center-based method shows stronger localization ability which is required for detecting small objects, while the anchor-based method can estimate the size of objects more precisely.

**Single-modality vs. multi-modality.** PointPainting [41] appends the segmentation scores to the input point clouds of CenterPoints [50], but the performance drops from 61.24% to 59.78%. We find

| Method | Vehicle | | | | Pedestrian | | | | Cyclist | | | | mAP |
|---|---|---|---|---|---|---|---|---|---|---|---|---|---|
| | overall | 0-30m | 30-50m | 50m-inf | overall | 0-30m | 30-50m | 50m-inf | overall | 0-30m | 30-50m | 50m-inf | |
| Multi-Modality (point clouds + images) | | | | | | | | | | | | | |
| PointPainting [41] | 66.46 | 83.70 | 56.89 | 40.74 | 47.62 | 58.95 | 39.33 | 23.34 | 65.27 | 73.48 | 61.53 | 43.90 | 59.78 |
| Single-Modality (point clouds only) | | | | | | | | | | | | | |
| PointRCNN [35] | 52.00 | 74.44 | 40.72 | 22.14 | 8.73 | 12.20 | 6.96 | 2.96 | 34.02 | 46.48 | 27.39 | 11.45 | 31.58 |
| PointPillars [23] | 69.52 | 84.51 | 60.55 | 45.72 | 17.28 | 20.21 | 15.06 | 11.48 | 49.63 | 60.15 | 42.43 | 27.73 | 45.47 |
| SECOND [47] | 69.71 | 86.96 | 60.22 | 43.02 | 26.09 | 30.52 | 24.63 | 14.19 | 59.92 | 70.54 | 54.89 | 34.34 | 51.90 |
| PV-RCNN [36] | 76.98 | 89.89 | 69.35 | 55.52 | 22.66 | 27.23 | 21.28 | 12.08 | 61.93 | 72.13 | 56.64 | 37.23 | 53.85 |
| CenterPoints [50] | 66.35 | 83.65 | 56.74 | 41.57 | 51.80 | 62.80 | 45.41 | 24.53 | 65.57 | 73.02 | 62.85 | 44.77 | 61.24 |

Table 5: Results of detection models on the testing split.

| Method | Vehicle | | | | Pedestrian | | | | Cyclist | | | | mAP |
|---|---|---|---|---|---|---|---|---|---|---|---|---|---|
| | overall | 0-30m | 30-50m | 50m-inf | overall | 0-30m | 30-50m | 50m-inf | overall | 0-30m | 30-50m | 50m-inf | |
| baseline [47] | 69.71 | 86.96 | 60.22 | 43.02 | 26.09 | 30.52 | 24.63 | 14.19 | 59.92 | 70.54 | 54.89 | 34.34 | 51.90 |
| $U_{small}$ | | | | | | | | | | | | | |
| BYOL [17] | 67.57 | 84.61 | 58.26 | 41.59 | 17.22 | 19.45 | 16.71 | 10.43 | 53.36 | 64.95 | 47.47 | 27.66 | 46.05 (-5.85) |
| PointContrast [46] | 71.53 | 87.02 | 62.37 | 47.23 | 22.68 | 26.33 | 21.58 | 12.98 | 58.04 | 70.01 | 51.74 | 31.69 | 50.75 (-1.15) |
| SwAV [6] | 72.25 | 87.20 | 63.38 | 48.93 | 25.11 | 29.32 | 23.50 | 14.13 | 60.67 | 70.90 | 55.91 | 35.39 | 52.68 (+0.78) |
| DeepCluster [5] | 72.06 | 87.09 | 63.09 | 48.78 | 27.56 | 32.21 | 26.60 | 13.61 | 50.30 | 70.33 | 55.82 | 35.89 | 53.31 (+1.41) |
| $U_{medium}$ | | | | | | | | | | | | | |
| BYOL [17] | 69.69 | 84.83 | 60.41 | 46.05 | 27.31 | 32.58 | 24.60 | 13.69 | 57.22 | 69.57 | 51.07 | 29.15 | 51.41 (-0.49) |
| PointContrast [46] | 70.15 | 86.71 | 61.12 | 48.11 | 29.23 | 35.52 | 36.28 | 13.06 | 58.91 | 70.05 | 53.86 | 34.27 | 52.76 (+0.86) |
| SwAV [6] | 72.10 | 87.11 | 63.15 | 48.58 | 28.00 | 33.10 | 25.88 | 14.19 | 60.17 | 70.46 | 55.61 | 34.84 | 53.42 (+1.52) |
| DeepCluster [5] | 72.12 | 87.31 | 62.97 | 48.55 | 30.06 | 36.07 | 27.23 | 13.47 | 60.45 | 70.81 | 54.93 | 36.03 | 54.21 (+2.31) |
| $U_{large}$ | | | | | | | | | | | | | |
| BYOL [17] | 72.23 | 87.30 | 63.13 | 48.31 | 23.62 | 27.10 | 22.14 | 13.47 | 60.45 | 70.82 | 55.31 | 35.65 | 52.10 (+0.20) |
| PointContrast [46] | 73.15 | 83.92 | 67.29 | 50.97 | 27.48 | 31.45 | 24.17 | 16.70 | 58.33 | 70.37 | 52.26 | 35.61 | 52.99 (+1.09) |
| SwAV [6] | 71.96 | 86.92 | 62.83 | 48.85 | 30.60 | 36.42 | 28.03 | 14.52 | 60.27 | 70.43 | 55.52 | 36.25 | 54.28 (+2.38) |
| DeepCluster [5] | 71.85 | 86.96 | 62.91 | 48.54 | 30.54 | 37.08 | 27.55 | 13.86 | 60.42 | 70.60 | 55.47 | 36.29 | 54.27 (+2.37) |

Table 6: Results of self-supervised learning methods on the testing split.

that the performance of PointPainting heavily relies on the accuracy of segmentation scores, and without explicit segmentation labels on the ONCE dataset, we cannot generate accurate semantic segmentation maps from images, which brings negative effects on 3D detection.

## 4.2 Self-Supervised Learning for 3D Object Detection

We reproduce 4 self-supervised learning methods, including 2 contrastive learning methods (Point-Contrast [46] and BYOL [17]), as well as 2 clustering-based methods (DeepCluster [5] and SwAV [6]) on our dataset. We choose the backbone of the SECOND detector [47] as the pretrained backbone. We first pretrain the backbone using self-supervised learning methods with different amounts of unlabeled data: 100k $U_{small}$, 500k $U_{medium}$ and 1 million $U_{large}$, and then we finetune the detection model on the training split. We report the detection performances on the testing split using the evaluation metric in appendix E. Performance on the validation split is also reported in appendix B. We provide training and implementation details in appendix C.

**Self-supervised learning on unlabeled data.** Experiments in Table 6 show that self-supervised methods can improve the detection results with enough unlabeled data. PointContrast [46] obtains 50.75% mAP with 100k unlabeled data, but the performance consistently improves to 52.76% and 52.99% with 500k and one million unlabeled data respectively, giving rise to 1.09% final performance gain over baseline. Self-supervised learning benefits from the increasing amount of unlabeled data.

**Contrastive learning vs. clustering.** The detection results indicate that clustering-based methods [5, 6] consistently outperforms contrastive learning methods [46, 17]. SwAV [6] and DeepCluster [5] achieve 54.28% and 54.27% mAP respectively on $U_{large}$, compared with 52.10% and 52.99% obtained by BYOL [17] and PointContrast [46]. This is mainly because constructing representative views of a 3D scene for contrastive learning is non-trivial in driving scenarios. Generating different views simply by performing different augmentations on the same point cloud may result in quite similar views that will make the pretraining process easily converge to a trivial solution.

| Method | Vehicle | | | | Pedestrian | | | | Cyclist | | | | mAP |
|---|---|---|---|---|---|---|---|---|---|---|---|---|---|
| | overall | 0-30m | 30-50m | 50m-inf | overall | 0-30m | 30-50m | 50m-inf | overall | 0-30m | 30-50m | 50m-inf | |
| baseline [47] | 69.71 | 86.96 | 60.22 | 43.02 | 26.09 | 30.52 | 24.63 | 14.19 | 59.92 | 70.54 | 54.89 | 34.34 | 51.90 |
| $U_{small}$ | | | | | | | | | | | | | |
| Pseudo Label [24] | 71.05 | 86.51 | 61.81 | 47.49 | 25.58 | 31.03 | 22.03 | 14.12 | 58.08 | 68.50 | 52.63 | 35.61 | 51.57 (-0.33) |
| Noisy Student [45] | 73.25 | 88.84 | 64.61 | 49.95 | 28.04 | 34.62 | 23.43 | 14.20 | 57.58 | 67.77 | 53.43 | 33.76 | 52.96 (+1.06) |
| Mean Teacher [40] | 74.13 | 89.34 | 65.28 | 50.91 | 31.66 | 37.44 | 29.90 | 14.61 | 62.69 | 71.88 | 59.22 | 39.45 | 56.16 (+4.26) |
| SESS [55] | 72.42 | 87.23 | 63.55 | 49.11 | 27.32 | 32.26 | 24.47 | 15.36 | 61.76 | 72.39 | 57.29 | 37.33 | 53.83 (+1.93) |
| 3DIoUMatch [42] | 72.12 | 87.05 | 63.65 | 50.35 | 31.41 | 38.56 | 27.62 | 14.25 | 59.46 | 69.53 | 54.82 | 36.18 | 54.33 (+2.43) |
| $U_{medium}$ | | | | | | | | | | | | | |
| Pseudo Label [24] | 70.72 | 86.21 | 61.72 | 47.39 | 21.74 | 25.73 | 19.91 | 13.28 | 56.01 | 67.14 | 50.18 | 33.23 | 49.49 (-2.41) |
| Noisy Student [45] | 73.97 | 89.09 | 65.35 | 51.04 | 30.32 | 36.24 | 27.08 | 16.24 | 61.35 | 71.28 | 56.70 | 37.96 | 55.22 (+3.32) |
| Mean Teacher [40] | 74.71 | 89.28 | 66.10 | 52.90 | 36.03 | 42.97 | 33.29 | 18.70 | 64.88 | 74.05 | 60.80 | 42.63 | 58.54 (+6.64) |
| SESS [55] | 72.60 | 87.02 | 64.29 | 50.68 | 35.23 | 42.59 | 31.40 | 16.64 | 64.67 | 73.93 | 61.14 | 40.80 | 57.50 (+5.60) |
| 3DIoUMatch [42] | 74.26 | 89.08 | 66.11 | 53.03 | 33.91 | 41.02 | 30.07 | 16.15 | 61.30 | 71.29 | 56.49 | 38.13 | 56.49 (+4.59) |
| $U_{large}$ | | | | | | | | | | | | | |
| Pseudo Label [24] | 70.29 | 85.94 | 61.18 | 46.66 | 21.85 | 25.83 | 20.22 | 12.75 | 55.72 | 66.96 | 50.29 | 32.92 | 49.29 (-2.61) |
| Noisy Student [45] | 74.50 | 89.23 | 67.11 | 53.15 | 33.28 | 40.20 | 28.89 | 17.50 | 62.05 | 71.76 | 57.53 | 39.32 | 56.61 (+4.71) |
| Mean Teacher [40] | 76.60 | 89.41 | 68.29 | 55.66 | 36.37 | 43.84 | 32.49 | 17.11 | 66.99 | 75.87 | 63.35 | 44.06 | 59.99 (+8.09) |
| SESS [55] | 74.52 | 88.97 | 66.32 | 52.47 | 36.29 | 43.53 | 33.15 | 16.68 | 65.52 | 74.63 | 62.67 | 41.91 | 58.78 (+6.88) |
| 3DIoUMatch [42] | 74.48 | 89.13 | 66.35 | 54.59 | 35.74 | 43.35 | 32.08 | 17.34 | 62.06 | 71.86 | 58.00 | 39.09 | 57.43 (+5.53) |

Table 7: Results of semi-supervised learning methods on the testing split.

## 4.3 Semi-Supervised Learning for 3D Object Detection

We implement 3 image-based semi-supervised methods: Pseudo Label [24], Mean Teacher [40] and Noisy Student [45], as well as 2 semi-supervised methods for point clouds in the indoor scenario: SESS [55] and 3DIoUMatch [42]. We first pretrain the model on the training split and then apply the 5 semi-supervised learning methods on both the training split and unlabeled scenes. We train those methods with 5 epochs for the 100k unlabeled subset $U_{small}$, 3 epochs for the 500k subset $U_{medium}$ and the 1 million subset $U_{large}$ during the semi-supervised learning process. We report detection performances on the testing split with the use of unlabeled subsets $U_{small}$, $U_{medium}$ and $U_{large}$ separately. Performance on the validation split is also reported in appendix B. We provide training and implementation details in appendix C.

**Semi-supervised learning on unlabeled data.** Experiments in Table 7 show that most semi-supervised methods can improve the detection results using unlabeled data. For instance, Mean Teacher [40] improves the baseline result by 8.09% in mAP using the largest unlabeled set $U_{large}$. The detection performance can be further boosted when the amount of unlabeled data increases. SESS [55] obtains 1.93% performance gain using 100k unlabeled scenes, and the performance gain reaches 5.60% with 500k unlabeled scenes and 6.88% with one million scenes.

**Pseudo labels vs. consistency.** There are two keys to the success of labeling-based methods [24, 45, 42]: augmentations and label quality. Without strong augmentations, the performance of Pseudo Label [24] drops from 51.90% to 49.29% albeit one million scenes $U_{large}$ are provided for training. 3DIoUMatch [42] adds additional step to filter out labels of low quality, and the performance reaches 57.43% compared with 56.61% of Noisy Student on $U_{large}$. Consistency-based methods [40, 55] generally perform better than labeling-based methods, and Mean Teacher obtains the highest performance 59.99% on $U_{large}$. SESS [55] performs worse than Mean Teacher with 58.78% mAP, which indicates that size and center consistency may not be useful in driving scenarios.

**Semi-supervised learning vs. self-supervised learning.** Our results in Table 6 and Table 7 demonstrate that the semi-supervised methods generally have a better performance compared with the self-supervised methods. Mean Teacher [40] attains the best performance of 59.99% mAP while the best self-supervised method SwAV [6] only obtains 54.28% on $U_{large}$. The major reason is that in semi-supervised learning the model usually receives stronger and more precise supervisory signals, *e.g.* labels or consistency with a trained model, when learning from the unlabeled data. However, in self-supervised learning, the supervisory signals on the unlabeled data are cluster id or similarity of pairs, which are typically noisy and uncertain.

## 4.4 Unsupervised Domain Adaptation for 3D Object Detection

Unsupervised domain adaptation for 3D object detection aims to adapt a detection model from the source dataset to the target dataset without supervisory signals on the target domain. Different datasets

| Task | Waymo → ONCE | | nuScenes → ONCE | | ONCE → KITTI | |
|------|------|------|------|------|------|------|
| Method | $AP_{BEV}$ | $AP_{3D}$ | $AP_{BEV}$ | $AP_{3D}$ | $AP_{BEV}$ | $AP_{3D}$ |
| Source Only | 65.55 | 32.88 | 46.85 | 23.74 | 42.01 | 12.11 |
| SN [43] | 67.97 (+2.42) | 38.25 (+5.67) | 62.47 (+15.62) | 29.53 (+5.79) | 48.12 (+6.11) | 21.12 (+9.01) |
| ST3D [49] | 68.05 (+2.50) | 48.34 (+15.46) | 42.53 (-4.32) | 17.52 (-6.22) | 86.89 (+44.88) | 41.42 (+29.31) |
| Oracle | 89.00 | 77.50 | 89.00 | 77.50 | 83.29 | 73.45 |

Table 8: Results on unsupervised domain adaptation. Source Only means trained on the source and directly evaluated on the target domain. Oracle means trained and tested both on the target domain.

typically have different collected environments, sensor locations and point cloud densities. In this paper, we reproduce 2 commonly-used methods: SN [43] and ST3D [49]. We follow the settings of those methods and conduct experiments on transferring the model trained on the nuScenes and Waymo Open dataset to our ONCE dataset, as well as transferring the model trained on the ONCE dataset to the KITTI dataset. The detection results of the car class are reported on the respective target validation or testing set using the KITTI $AP_{3D}$ metric following [43, 49].

**Statistical normalization vs. self-training.** The normalization-based method SN [43] surpasses the Source Only model by $2.42\%$ in $AP_{BEV}$ and $5.67\%$ in $AP_{3D}$ on the Waymo → ONCE adaptation task, and the self-training method ST3D [49] also attains a considerable performance gain with $15.46\%$ $AP_{3D}$ improvement. However, ST3D performs poorly on the nuScenes → ONCE task. It is mainly because the nuScenes dataset has fewer LiDAR beams, and the model trained on nuScenes may produce more low-quality pseudo labels, which will harm the self-training process of ST3D. Although those two methods achieve strong results on the adaptation from/to the ONCE dataset, there is still a gap with the Oracle results, leaving large space for future research.

## 5    Conclusion

In this paper, we introduce the ONCE (**O**ne millio**N** s**C**en**E**s) dataset, which is the largest autonomous driving dataset to date. To facilitate future research on 3D object detection, we additionally provide a benchmark for detection models and methods of self-supervised learning, semi-supervised learning and unsupervised domain adaptation. For future works, we plan to support more tasks on autonomous driving, including 2D object detection, 3D semantic segmentation and planning.

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
