# OpenReview forum: "One Million Scenes for Autonomous Driving: ONCE Dataset"
_NeurIPS.cc/2021/Track/Datasets_and_Benchmarks/Round1 — NeurIPS 2021 Datasets and Benchmarks Track (Round 1)_

### Official Review · Reviewer_4AUR · 2021-06-22
**A useful large-scale dataset for 3D autonomous driving with diverse scenes**

**Rating:** 7
**Confidence:** 2
**Clarity:** This paper is well written and organi…

**Strengths:**

+ The authors contribute the largest dataset for 3D object detection in autonomous driving scenario, which requires a lot of efforts.
+ Compared with existing datasets, ONCE contains more scenes, images, larger size and area.
+ The dataset is released.
+ Various evaluations have been done on many existing detection models. I believe this will be beneficial for future research and comparison.
+ This paper is well written and organized.


**Weaknesses:**

- Table.1 shows the comparison of different datasets, which is good. But I think some important information is missing. For example, it is good to see what the aim is for each dataset, self-supervised/semi-supervised/unsupervised.
- The number of annotations is actually less then that of Waymo Open and nuScenes, which limits its usage if one wants to have more training data with annotations, given the large scale of ONCE.
- L166-174, it is not clear about the dataset splits. For example, how do you divide the dataset for each split? What is the information for weather, time and areas in each split? If it needs too much space, the author can put it in Supp. material.


**Additional Feedback:**

- It will be great to see a new model proposed on this dataset. Especially, one of the goal of ONCE is to facilitate research of unsupervised detection. It will be interesting to see one.

- How do you guarantee the quality of annotation?



**Correctness:**

The dataset is constructed in a sound way. The evaluation and experiment sound reasonable to me.

**Documentation:**

The documentation is good. A webpage is accompanied with the proposed dataset for future research.

**Ethics:**

I don't find any ethics problem, given the data protection problem is resolved (L129-133).

**Relation To Prior Work:**

Related works have been sufficiently and clearly discussed. References in this paper are appropriate.

**Summary And Contributions:**

This paper introduces a large-scale dataset, ONCE, for 3D object detection in autonomous driving scenario. The proposed dataset consists of 1 million LiDAR scenes and 7 million corresponding camera images. Further, ONCE is constructed from 144 driving hours with various areas, periods and weather conditions, which makes the proposed one larger and more challenging than Waymo Open, the existing largest dataset for 3D detection. Moreover, this paper introduces different evaluation protocols for self-supervised, semi-supervised and unsupervised methods. Various models have been benchmarked on this dataset for future comparison.

---

> ### Author Response · Authors · 2021-07-08
> **Thanks for your comments**
>
> **Q1: Table.1 shows the comparison of different datasets, which is good. But I think some important information is missing. For example, it is good to see what the aim is for each dataset, self-supervised/semi-supervised/unsupervised.**
>
> Existing datasets in Table.1 mainly focus on the task of supervised 3D object detection. To the best of our knowledge, our ONCE dataset is the first work that provides a data source and benchmark for self-supervised and semi-supervised 3D object detection. For unsupervised domain adaptation, the commonly used existing datasets are KITTI, Waymo, nuScenes and Lyft.
>
> **Q2: The number of annotations is actually less than that of Waymo Open and nuScenes, which limits its usage if one wants to have more training data with annotations, given the large scale of ONCE.**
>
> Please refer to C1 for the clarifications.
>
> **Q3: L166-174, it is not clear about the dataset splits. For example, how do you divide the dataset for each split? What is the information for weather, time and areas in each split? If it needs too much space, the author can put it in Supp. material.**
>
> Thanks for your advice. Please refer to C3 and Sec. 3.2 in the paper for details.
>
> **Q4: It will be great to see a new model proposed on this dataset. Especially, one of the goal of ONCE is to facilitate research of unsupervised detection. It will be interesting to see one.**
>
> This track mainly aims to benchmark existing methods so at this time we didn’t propose a new model on unsupervised detection, but we plan to explore this area and propose novel methods as future works.
>
> **Q5: How do you guarantee the quality of annotation?**
>
> We use images to calibrate the annotated 3D boxes in point clouds and the annotations will go through a double-check process for refinement. Please refer to C3 and Sec. 3.1 in the paper for details.

---

> > ### Comment · Reviewer_4AUR · 2021-07-19
> > **Reply to authors' reply**
> >
> > Thank you very much for your efforts in addressing my concerns. Overall, I think this work can bring some interest to the community. I don't have further questions. Since I am not an expert in this field. I will keep my score unchanged.

---

### Official Review · Reviewer_2ZJF · 2021-07-01
**ONCE Dataset Review**

**Rating:** 7
**Confidence:** 4
**Correctness:** Yes, the evaluation seems to be correct.
**Clarity:** Yes, the paper is clear.

**Strengths:**

+ ONCE contains more driving scenes and more hours than the current datasets.
+ ONCE also contains data from different geographical location, weathers and times.
+ Backbone trained on ONCE is superior compared to the ones trained on Waymo and nuScenes on the downstream KITTI dataset.
+ Insights on use of pixels/voxels/pillars and anchor/center assignments are nice.
+ Bench-marking of self-supervised and semi-supervised methods is nice.

**Weaknesses:**

- ONCE has five categories. This is much less compared to the nuScenes dataset, which contains 23. Even in evaluation, nuScenes uses ten categories (way more than ONCE).

- The supervised benchmark of ONCE is small compared to other benchmarks. nuScenes has 40k images, Waymo Open has 200k images, while ONCE only has 16k. The train + Val split is of 8k images, which is similar to KITTI. Although I agree that the number of images is more compared to nuScenes and Waymo, I doubt that the supervised learning community would use the ONCE dataset for benchmarking their models.

- Comparison of diversity missing. It is worthwhile to compare the diversity of ONCE, NuScenes, Waymo, and BDD100k (2D object detection) to get a sense of how ONCE is more diverse than other datasets. In other words, a histogram plot of weather, times, and areas instead of a pie chart comparing the four datasets in Figure 3 would be nice.

- BDD dataset [1] has more RGB images (120m) compared to ONCE. Although it does not have 3D labels, BDD100k has a task for lane detection, semantic segmentation, instance segmentation, and segmentation. A more reasonable alternative to release the ONCE dataset is to annotate 3D labels on the BDD 100k dataset rather than coming up with a new dataset altogether.  Annotating BDD100K also makes sense as BDD100k has 1.8m objects compared to the 417k objects in ONCE.

- An additional feature of the nuScenes dataset (against Waymo dataset) is nuScenes also contains 3D annotations for both RGB and Lidar maps. Does ONCE provide the 3D annotations for RGB images as well?

- Does ONCE have multi-lane data? It is not super clear to me. nuScenes does have multi-lane data.

References-
[1] BDD100K, Yu et al , CVPR 2020.

**Additional Feedback:**

- In Implementation Details (Section C of the Appendix), please provide the details of data augmentation, learning rate policy and optimizer of each of the methods. It is also worthwhile to put the data split of each of the benchmark for clarity.

**Documentation:**

There are all sufficient details in the appendix. URL is also present.

**Ethics:**

No, I do not think there are concerns.

**Relation To Prior Work:**

Somewhat unclear to me.

**Summary And Contributions:**

The paper introduces the ONCE dataset which contains 1m LiDAR scenes and 7m camera images from 144 driving hours. The dataset is collected across different geographical locations, weather and times.  The paper shows that backbone trained on ONCE is superior compared to the ones trained on Waymo and nuScenes on the downstream KITTI dataset. The paper additionally benchmarks a bunch of self-supervised and semi-supervised methods on the ONCE dataset.

---

> ### Author Response · Authors · 2021-07-08
> **Thanks for your comments**
>
> **Q1: ONCE has five categories. This is much less compared to the nuScenes dataset, which contains 23. Even in evaluation, nuScenes uses ten categories (way more than ONCE).**
>
> Our 5 categories (Car/Bus/Truck/Pedestrian/Cyclist) have already covered most objects in driving scenarios. NuScenes provides 10 fine-grained categories but most of those categories have been covered by our definition. For example, the Truck class on ONCE equals to Truck/Trailer/Construction Vehicle on nuScenes, and the Cyclist class on ONCE equals to Motorcycle/Bicycle on nuScenes. We note that Waymo also has only 4 categories (Vehicle/Pedestrian/Cyclist/Sign) and KITTI has only 3 categories (Car/Pedestrian/Cyclist).
>
> **Q2: The supervised benchmark of ONCE is small compared to other benchmarks. nuScenes has 40k images, Waymo Open has 200k images, while ONCE only has 16k. The train + Val split is of 8k images, which is similar to KITTI. Although I agree that the number of images is more compared to nuScenes and Waymo, I doubt that the supervised learning community would use the ONCE dataset for benchmarking their models.**
>
> Please refer to C1 for the clarifications.
>
> **Q3: Comparison of diversity missing. It is worthwhile to compare the diversity of ONCE, NuScenes, Waymo, and BDD100k (2D object detection) to get a sense of how ONCE is more diverse than other datasets. In other words, a histogram plot of weather, times, and areas instead of a pie chart comparing the four datasets in Figure 3 would be nice.**
>
> Please refer to C2 and Sec. 3.3 in the paper for the diversity comparison. The comparison is conducted on Waymo, nuScenes and ONCE since all those 3 datasets contain 3D point clouds and focus on 3D object detection. BDD100k is a wonderful dataset but it focuses on 2D detection and segmentation which is different from Waymo/nuScenes/ONCE. Thus we don’t think it’s necessary to compare ONCE with BDD.
>
> **Q4: BDD dataset [1] has more RGB images (120m) compared to ONCE. Although it does not have 3D labels, BDD100k has a task for lane detection, semantic segmentation, instance segmentation, and segmentation. A more reasonable alternative to release the ONCE dataset is to annotate 3D labels on the BDD 100k dataset rather than coming up with a new dataset altogether. Annotating BDD100K also makes sense as BDD100k has 1.8m objects compared to the 417k objects in ONCE.**
>
> BDD is a wonderful dataset for multiple 2D tasks. However, compared to the 3D autonomous driving datasets which typically contain both images and point clouds, BDD only contains images. And without 3D point clouds, it is not feasible to directly annotate 3D labels on the BDD dataset, since we cannot obtain the accurate depth information of each box on images. 3D labels are generally annotated in 3D point clouds and 2D images are only used to calibrate the size of 3D boxes. On the other hand, compared to annotating BDD, our ONCE dataset also has its own merits since we provide a new large-scale and diverse 3D data source with one million 3D point clouds which BDD doesn’t have.
>
> **Q5: An additional feature of the nuScenes dataset (against Waymo dataset) is nuScenes also contains 3D annotations for both RGB and Lidar maps. Does ONCE provide the 3D annotations for RGB images as well?**
>
> Yes. Please refer to Fig. 4 in the appendix. We provide 3D annotations for RGB images by projecting those 3D annotations in point clouds onto the image planes.
>
> **Q6: Does ONCE have multi-lane data? It is not super clear to me. nuScenes does have multi-lane data.**
>
> Currently our ONCE dataset only focuses on the task of 3D object detection. Multi-lane data is not necessary for 3D object detection so we didn’t provide this data.
>
> **Q7: In Implementation Details (Section C of the Appendix), please provide the details of data augmentation, learning rate policy and optimizer of each of the methods. It is also worthwhile to put the data split of each of the benchmark for clarity.**
>
> Thanks for your advice. We have updated this part. Please refer to appendix C for more details.

---

### Official Review · Reviewer_eUVP · 2021-07-06
**A large-scale dataset for autonomous driving**

**Rating:** 6
**Confidence:** 4
**Clarity:** Yes

**Strengths:**

This dataset shows that unlabeled data which are relatively cheap can improve the network’s performance on important downstream tasks where annotations can be expensive to collect for autonomous driving. I think this would motivate future research on how to collect even better or more diverse unlabeled 3D data for better pre-training networks, which improves the reliability of autonomous driving.

**Weaknesses:**

It contains very few annotations of 3D boxes compared to existing datasets such as nuScenes and Waymo. The authors show the diversity of the dataset but it is unclear how it compares with existing datasets.

**Additional Feedback:**

The authors should put table 4 in their appendix which shows how pre-training on ONCE improves performance on nuScenes and Waymo in the main paper. This is an important experiment and I almost missed it because it was in the appendix.

**Correctness:**

They construct this dataset by attaching LiDAR sensors and cameras on a car. The claims in the submission seem to be correct.

**Documentation:**

I think some details are missing on data collection. It is unclear how the 3D annotations are obtained, how the authors choose the scenes for annotation and where they collect the data.

**Ethics:**

No. The authors blur objects that may contain personal information in the dataset.

**Relation To Prior Work:**

Yes

**Summary And Contributions:**

This paper proposes a new large-scale dataset for autonomous driving. The new dataset consists of nearly 1 million unlabeled scenes and 16k annotated scenes which in total provide 1 million point clouds and 7 million images. The annotated scenes include 3D bounding boxes of 5 categories including car, bus, truck, pedestrian and cyclist. It also extends the existing evaluation metric for 3D object detection by also taking the object orientations into account. Experiments show that pre-training on the unlabeled data improves the performance of the network on the 3D object detection tasks.

---

> ### Author Response · Authors · 2021-07-08
> **Thanks for your comments**
>
> **Q1: It contains very few annotations of 3D boxes compared to existing datasets such as nuScenes and Waymo.**
>
> Please refer to C1 for detailed explanations.
>
> **Q2: The authors show the diversity of the dataset but it is unclear how it compares with existing datasets.**
>
> Please refer to C2 and Sec. 3.3 in the paper for the diversity comparison.
>
> **Q3: I think some details are missing on data collection. It is unclear how the 3D annotations are obtained, how the authors choose the scenes for annotation, and where they collect the data.**
>
> Please refer to C3 and Sec. 3.1 in the paper for detailed data collection and annotation.
>
> **Q4: The authors should put table 4 in their appendix which shows how pre-training on ONCE improves performance on nuScenes and Waymo in the main paper. This is an important experiment and I almost missed it because it was in the appendix.**
>
> Thanks for your advice. We have put this table in the main paper.

---

### Author Response · Authors · 2021-07-08
**Paper updates and clarifications**

We have updated the main paper and appendix and highlighted the modified parts. We added quality analysis and comparisons (Table 3) and diversity analysis and comparisons (Table 4) to the main paper. We added a short description of the data collection and annotation process in Sec.3.1 and detailed dataset splits in Sec.3.2. We added more implementation details and an example of 3D annotations on an RGB image in the appendix.

We thank all the reviewers for their constructive comments. We still want to make some clarifications on the frequently asked questions:

**Q1: The number of annotations is less than that of Waymo and nuScenes, which will restrict its potential usage to benchmark the supervised learning models.**

**C1:** We want to explain this problem from 3 aspects.

(1) The goal of our ONCE dataset is different from that of the Waymo and nuScenes dataset. As we mentioned in the abstract and L.28-L.43, given the fact that current perception models greatly rely on massive annotated data while the 3D annotation process is relatively time-consuming, how to leverage fewer labels and more unlabeled data becomes a practical problem for self-driving. Thus our goal is not to build a dataset that has more annotations than nuScenes and Waymo, but to provide a platform that with large-scale, real-world, and diverse scenes, as well as a sufficient amount of labeled data to train a decent 3D detector, researchers can explore methods to further improve the generalizability and accuracy of detectors by leveraging the unlabeled scenes. From this aspect, we believe our ONCE dataset, which has the existing largest number of scenes and a sufficient amount of labeled data, has already supported our claim.

(2) The amount of labeled data is sufficient to support research on supervised 3D detection and benchmark different supervised models. We admit that large-scale labeled data like Waymo and nuScenes can improve detection performance. However, if the goal is to develop, or to benchmark supervised 3D detection models, the labeled data amount provided by ONCE is completely sufficient. This is because the results of supervised models are always consistent (different in scores, but similar in rankings) across different amounts of labeled data. For example, the performance comparisons of different supervised 3D models are generally consistent across KITTI, Waymo, and nuScenes, although KITTI has much fewer annotations than nuScenes and Waymo. The labeled amount of our ONCE dataset is even larger than that of KITTI with 360 degrees of annotations compared to only FOV annotations, and the results of ONCE supervised benchmark (Sec. 4.1) are also consistent with those results on KITTI, Waymo, and nuScenes. Thus we believe the labeled set of ONCE is sufficient for developing and benchmarking supervised models.

(3) The number of annotated boxes is not the only factor considered for supervised learning. It is worth noting that the boxes of our ONCE dataset are annotated at the rate of 1 FPS, which guarantees a higher coverage of diverse scenes. Although Waymo provides 230k annotated frames, their annotation rate is 10 FPS, and most adjacent frames are quite similar thus redundant. Practical experiments show that detectors trained on 20% of Waymo data (2 FPS, 46k scenes) can achieve almost the same performance with 100% data. Thus if you consider the actually used annotated scenes (46k of Waymo, 40k of nuScenes compared to 16k of ONCE), the difference of labeled amounts is not that large.

**Q2: Questions on the diversity analysis compared to other datasets.**

**C2:** We have updated our paper and added diversity comparisons (Table 4) in the main paper. Please refer to the latest version. We note that the quality comparisons (Table 3) also indicate that our ONCE dataset is more diverse than Waymo and nuScenes. The backbone pretrained on ONCE shows superior performance on the downstream tasks, compared to the backbones pretrained on Waymo and nuScenes. This experiment indicates that the data of ONCE can be more diverse than that of Waymo and nuScenes.

**Q3: Questions on the dataset splits, the data collection, and the annotation process.**

**C3:** We have added more details on the dataset splits (Sec. 3.2), the data collection and annotation process (Sec. 3.1) to the main paper. Please refer to the latest version for details.

---

> ### Comment · Reviewer_2ZJF · 2021-07-12
> **Reply to Authors' Response**
>
> Thank you, authors, for your response and for adding all the relevant information to the paper. I agree that ONCE covers most of the categories seen frequently in autonomous driving. I also agree that annotating BDD100k is not possible as they do not have point cloud information. Thank you for the clarification that ONCE has boxes for RGB images, does not have multi-lane data and also including the data split and other implementation details.
>
> **The number of annotated boxes is not the only factor considered for supervised learning. It is worth noting that the boxes of our ONCE dataset are annotated at the rate of 1 FPS, which guarantees a higher coverage of diverse scenes. Although Waymo provides 230k annotated frames, their annotation rate is 10 FPS, and most adjacent frames are quite similar thus redundant.**
>
> Let us divide total images by fps then for ONCE, Waymo and nuScenes. Then, Total images/fps for Waymo open is ~23k, while for ONCE is 16k. Can we have the fps and images/fps for nuScenes as well?
>
> **Diversity not compared with BDD100k**
>
> I agree with the authors that BDD100k is a 2D detection dataset. However, I disagree with the authors that the comparison with BDD100k is futile. BDD100k is one of the most diverse datasets for **RGB** object detection. Therefore, it is good to compare with BDD100k for the sake of completeness to the readers. If ONCE is more diverse, the readers know that ONCE is a great dataset to use. And if ONCE is not that diverse as BDD100k, the readers know that the next lidar dataset should be at least as diverse as BDD100k.

---

> > ### Author Response · Authors · 2021-07-13
> > **Thanks for the comments**
> >
> > **Q4: Detailed comparisons on the annotation amount of nuScenes, Waymo, and ONCE.**
> >
> > C4: nuScenes has 400k frames of lidar data with a 20Hz collection rate. 40k frames are annotated at the rate of 2Hz. Waymo has 230k frames of lidar data with a 10Hz collection rate. All frames are annotated, but most researchers use 46k frames (2Hz) in practice due to the redundancy. ONCE collect 1M lidar data at the rate of 10 Hz. 16k frames are selected and annotated at the rate of 1Hz.
> >
> > We admit that ONCE has fewer annotations than nuScenes/Waymo. However, the difference of the labeled amount is not large, and the amount of labeled data on ONCE is sufficient for fully/semi/self-supervised learning on 3D object detection. Please refer to C1 for details.
> >
> > **Q5: Diversity comparisons with BDD100k.**
> >
> > C5: We added a comparison with BDD100k in Table 4 of the main paper. ONCE has a similar level of diversity compared to BDD100k. ONCE has 4 types of labels on time periods while BDD only has 3 labels. Both two datasets contain sufficient scenes in daytime and night. BDD has more weather types (e.g. snowy and foggy) but this is not significant for 3D object detection, since 3D detection is mainly conducted on point clouds that are not sensitive to snowy and foggy weather. And ONCE also contains sufficient scenes on sunny/cloudy/rainy days. BDD has 6 types of area labels and all road types have been covered by ONCE, while ONCE provides 1.02% of scenes on bridges that BDD doesn’t have.
> >
> > In conclusion, ONCE and BDD have completely different goals (ONCE is built for fully/semi/self-supervised learning on 3D detection while BDD is for 2D detection), different data modalities (ONCE contains well-aligned point clouds and images while BDD contains only images), and similar diversity levels (both contains fine-grained labels for all kinds of driving time, weather and areas). We believe ONCE and its corresponding benchmark can be considered as a technical contribution to the 3D autonomous driving community.

---

### Decision · Program_Chairs · 2021-07-26

**Decision:**

Accept

**Comment:**

All reviewers recommended acceptance. The AC concurs.